# Involvement of IL-33 in the Pathophysiology of Systemic Lupus Erythematosus: Review

**DOI:** 10.3390/ijms23063138

**Published:** 2022-03-15

**Authors:** Julie Sarrand, Muhammad Soyfoo

**Affiliations:** Department of Rheumatology, Hôpital Erasme, Université Libre de Bruxelles, 1070 Brussels, Belgium; julie.sarrand@ulb.be

**Keywords:** systemic lupus erythematosus, IL-33, cytokines, alarmins, autoimmune disease, innate immunity, inflammation

## Abstract

IL-33 is a newly discovered cytokine displaying pleiotropic localizations and functions. More specifically, it also functions as an alarmin, following its release from cells undergoing cell death or necrosis, to alert the innate immune system. The role of IL-33 has been underlined in several inflammatory and autoimmune diseases including systemic lupus erythematosus (SLE). The expressions of IL-33 as well as its receptor, ST2, are significantly upregulated in SLE patients and in patients with lupus nephritis. This review discusses the involvement of IL-33 in the pathology of SLE.

## 1. Introduction 

Systemic lupus erythematosus (SLE) is the prototype of autoimmune connective tissue disease and affects mainly young women of childbearing age [1]. It is characterized by chronic and aberrant immune activation against self-antigens, ultimately leading to the production of autoantibodies and immune complexes (ICs) entailing further damage in multiple organs such as the joints, skin, kidneys, lungs and brain [2]. Even though the contribution of the innate and adaptive immune systems to the break of tolerance towards autoantigens is well established, the exact mechanisms underlying this phenomenon still remain elusive. 

In genetically predisposed individuals exposed to a wide range of environmental factors [3], products of cell damage are potent activators of endosomal Toll-like receptors (TLR) and TLR-independent nucleic acid sensors expressed by innate immune cells such as dendritic cells (DCs). Specifically, TLR-7 and TLR-9 are, respectively, activated by single-stranded ribonucleic acid (RNA) and unmethylated deoxyribonucleic acid (DNA) found in products of cell damage and ICs, further leading to strong type I interferon (IFN) production [4,5]. This goes in line with the so-called “IFN gene signature” observed in peripheral blood mononuclear cells (PBMCs) of SLE patients [6,7]. More recently, NETosis has also been incriminated in the pathophysiology of SLE. Neutrophil extracellular traps (NETs) are a fibrous network extruded by activated neutrophils primarily composed of DNA and pro-inflammatory proteins. Studies conducted in SLE patients revealed that NETs induce the production of type I IFN by DCs, serve as self-antigens for presentation to T lymphocytes and mediate vascular damage and thrombosis [8].

In addition, type I IFN induces the activation of antigen-presenting DCs and drastically increases their capacity to present autoantigens released from dying cells to T cells. The ensuing generation of T effector cells results in the production of inflammatory cytokines and the sustained expression of the cluster of differentiation (CD)40 ligand (CD40L) that supports the activation of autoreactive B cells [9], which further leads to autoantibody production, a hallmark of SLE. In addition, T cells of SLE patients are characterized by a decrease in interleukin (IL)-2 production, which reduces the production of regulatory T cells (Treg) [10]. Moreover, some studies have shown an increased number of type 17 helper (Th17) cells together with increased levels of IL-17 in patients with lupus nephritis [11].

The production of autoreactive B cells leads to the secretion of pathogenic autoantibodies, further perpetuating inflammation and organ damage by IC deposition (containing nucleic acids, nucleic acid-binding proteins and autoantibodies directed against those components) and complement and neutrophil activation. Cell debris emerging from cell damage results in the production of type I IFN and other pro-inflammatory mediators and further triggers the activation of innate immunity [12]. Furthermore, functional inactivation of autoreactive B cells fails to eliminate autoreactive B cells from SLE patients [13]. Autoreactive B cells are very efficient antigen-presenting cells and potent activators of T cells. This results in a phenomenon of cross-activation where both B and T cells can activate each other, leading to the phenomenon of epitope spreading, reinforcing the loop of autoimmunity [14]. 

IL-33 is a cytokine that was first identified approximately 20 years ago as a ligand for the IL-1 receptor (IL-1R) family member suppression of tumorigenicity 2 (ST2) [15], and it has been associated with several biological processes and plays a pivotal role in innate and adaptive immunity, tissue repair, homeostasis and responses to environmental stresses. IL-33 is believed to act as an alarmin, as it is passively released by damaged or necrotic barrier cells (endothelial and epithelial cells) [16]. Alarmins mediate intercellular signals through interactions with chemotactic and pattern recognition receptors (PRRs) to foster innate immune cells. Additionally, alarmins have the ability to elicit adaptive immunity responses and T cell-dependent long-term immune memory through their capacity to induce DC maturation [17]. IL-33 primarily induces type 2 helper (Th2) immune responses through its receptor ST2 [18]. However, recent studies found ST2 expression on Th1 cells, Treg cells, group 2 innate lymphoid cells (ILC2), CD8+ T cells and natural killer (NK) cells [19,20].

Much current interest in IL-33 has been prompted by its role in several inflammatory and autoimmune diseases including SLE, Sjögren’s syndrome, systemic sclerosis and rheumatoid arthritis [21,22,23,24]. However, the contribution of the IL-33/ST2 axis to the pathogenesis of SLE still remains incompletely defined. 

In the present review, we aim to depict the current state of knowledge regarding the involvement of the IL-33/ST2 axis in the pathogenesis of SLE.

## 2. IL-33 and ST2: Biology and Functions

IL-33 is a member of the IL-1 family cytokines that encompass IL-1, IL-18 and IL-36 [25] and is constitutively expressed in the nucleus of non-immune cells, more particularly in endothelial and epithelial cells, fibroblasts and myofibroblasts [26,27]. Upon physiological conditions, IL-33 is localized in the nucleus, bound to chromatin (via the tails of histones H2A and H2B) [28], and acts as the keeper of epithelial barrier integrity through its transcriptional regulation abilities [29,30,31].

In pathological settings, if a breach in the epithelial barrier occurs, ensuing mechanical stress-induced cell death or necrosis, IL-33 is passively released in the extracellular compartment where it acts as an alarmin or damage-associated molecular pattern (DAMP) [32]. Extracellular full-length IL-33 is processed by proteases derived from neutrophils [33] and mast cells [34], generating truncated forms displaying biological activity up to 30-fold higher than the full-length IL-33 [33,34]. Extracellular IL-33 exerts its functions through the receptor ST2 and its coreceptor IL-1 receptor accessory protein (IL-1RacP, also known as IL1-R3). Due to alternative splicing, three isoforms of ST2 have been described: the transmembrane receptor type (ST2L), the soluble form (sST2) and the variant ST2 (ST2V) [35,36,37].

The binding of IL-33 to the transmembrane receptor ST2 enables its dimerization with IL-1RacP, further activating intracellular signaling through the myeloid differentiation primary response 88 (MyD88) adaptor, interleukin receptor-associated kinase (IRAK)1, IRAK4 and tumor necrosis factor receptor-associated factor (TRAF)6. This enables the activation of mitogen-activated protein (MAP) kinases and the nuclear factor κB (NFκB) transcription factor, leading to cell proliferation and the secretion of pro-inflammatory cytokines such as IL-4, IL-5 and IL-13 [38,39] (Figure 1).

The IL-33/ST2 axis is tightly regulated at several levels. sST2 acts as a decoy receptor of IL-33 and prevents its interaction with ST2, thereby counteracting its systemic effects [40]. The IL-33/ST2 axis is also antagonized by the single immunoglobulin domain IL-1R-related molecule (SIGGIR; also known as TIR8) that splits the heterodimer ST2/IL-1RacP, and by the activation of the ubiquitin–proteasome system, which digests ST2 [41,42]. Once released in the extracellular environment, inactivation of IL-33 occurs rapidly after approximatively 2 h through the oxidation of its cysteine residues and the formation of disulfide bridges [43].

The IL-33/ST2 axis mediates the activation of both myeloid and lymphoid cells and induces mainly a type 2 immune response, through the secretion of Th2-type cytokines (IL-5 and IL-13) and Th2 cell polarization [44,45]. ST2 is found in a wide variety of immune cells, including mast cells [46], basophils [47], eosinophils [48], M2 macrophages [49], neutrophils [47], NK cells [19], innate NK (iNK) cells [19], ILC2 [50], Treg [51] and Th2 cells [44]. However, under specific circumstances, IL-33 can also promote type 1 and type 17 immune responses [52]. More specifically, the production of type I IFN has been demonstrated following IL-33 exposure, leading to ST2 activation in type 1 helper (Th1) cells, NK cells and CD8+ T cells [19,20,53]. In asthma mouse models, the IL-33/ST2 activation in mast cells triggered a Th17 immune response [53]. In vitro studies on mouse macrophages showed that IL-33 exposure increased the expression of TLR-4, myeloid differentiation protein (MD)-2 and MyD88 [54]. In mouse bone marrow-derived DCs, the activation of ST2 upon IL-33 exposure increased the expression of DC maturation markers (CD80, CD40), pro-inflammatory cytokines (IL-4, IL-5, IL-13, tumor necrosis factor (TNF)-α and IL-1β) and chemokines such as C-C motif chemokine ligand 17 (CCL17) [55]. Therefore, IL-33 is a potent initiator of the innate immune response and can further activate adaptive immunity. 

## 3. The Role of IL-33/ST2 Axis in Inflammatory Diseases

A growing body of evidence indicates that the IL-33/ST2 axis exerts a dichotomous role in inflammatory diseases, ensuing protective or deleterious effects, depending on the immune context. In the last decade, studies have identified IL-33 as a potential culprit in numerous inflammatory diseases such as asthma [56], inflammatory bowel disease (IBD) [57], rheumatoid arthritis [58], systemic sclerosis [59] and systemic lupus erythematosus (SLE) [60]. For instance, IL-33 and ST2 levels are significantly increased in both the serum and synovium of patients with rheumatoid arthritis, the archetype of inflammatory rheumatic diseases, and even correlated with disease activity [30]. In mouse models of rheumatoid arthritis, IL-33 administration worsened the disease pattern, whereas antagonizing IL-33 signaling significantly decreased disease activity [61,62]. 

However, consistent evidence has also pled a more protective effect of the IL-33/ST2 axis, particularly in mucosal healing processes in IBD [63], immunosuppression in severe sepsis [64] and the reduction in the formation of atherosclerotic plaques in cardiovascular diseases [65]. For example, recombinant IL-33 treatment alleviated colitis in mouse models of Crohn’s disease. This was closely linked to a switch from Th1 toward Th2 and Treg cells [63].

When released from epithelial cells following cell death or mechanical stress, IL-33 acts on various immune cells via its ST2 receptor and elicits both innate and adaptive immune responses. IL-33 induces innate immunity through activation of innate immune cells such as mast cells [66], basophils [67], eosinophils [67] and ILC2 [68], leading to the secretion of type 2 pro-inflammatory cytokines (i.e., IL-4, IL-5 and IL-13). The activation of DCs entails the polarization of naïve CD4+ T cells to a Th2 phenotype [69]. Under certain conditions, IL-33 can also support a type 1 cytokine response through the activation of Th1 cells, cytotoxic T lymphocytes and NK cells, thereby explaining the capacity of these cells to produce type I IFN upon IL-33 exposure [19,70,71].

The anti-inflammatory functions of IL-33 are mediated by Treg cells [72], regulatory B (Breg) cells [70] and M2 macrophages [71]. IL-33 supports—directly and indirectly—Treg cell proliferation, through the secretion of IL-2 by innate cells such as DCs [73] and mast cells [74]. Furthermore, Treg secrete amphiregulin (AREG), an epidermal growth factor receptor ligand that supports tissue repair [75,76]. IL-33 is a potent inducer of IL-10-producing Breg, which confers effective protection against mucosal inflammatory disorders in mice [70]. Besides supporting a type 2 immune response, ILC2 activation by IL-33 promotes tissue repair via the secretion of AREG [76,77] and induces the generation of M2-polarized macrophages [71] (Figure 2).

## 4. Expression of IL-33 and ST2 in Systemic Lupus Erythematosus

The IL-33/ST2 axis has been recently incriminated in the pathogenesis of SLE, but its precise contribution still remains elusive, partly due to the lack of clinical studies. 

The human IL-33 gene is located on chromosome 9p24.1 in humans [78]. The association between IL-33 gene polymorphisms and SLE has been studied exclusively in the Chinese population. Two polymorphisms, the rs1929992-G and rs1891385-C alleles, have been linked to the risk of SLE [79,80,81]. However, the increase was only moderate, with an odds ratio of 1.4 to 1.6 for the rs1891385C allele [79,80], and 1.3 to 1.6 for the rs1929992-G allele [80,81]. In addition, IL-33 serum levels of SLE patients only correlated with the rs1891385C allele [79]. 

Conflicting data exist regarding the serum levels of IL-33 in SLE patients. It was reported in several studies that IL-33 levels were significantly increased in the serum of patients with SLE compared with healthy controls [21,79,82,83,84]. IL-33 levels correlated with the disease activity score (Systemic Lupus Erythematosus Disease Activity Index, SLEDAI) [84] and acute inflammatory parameters such as the erythrocyte sedimentation rate (ESR) and C reactive protein (CRP), suggesting a potential interest for its use as a surrogate marker in the acute phase of SLE [82]. In addition, a study found increased amounts of extracellular IL-33 complexed with NETs in blood, skin and kidney biopsies from SLE patients, which correlated with the disease activity. Ex vivo analysis confirmed that neutrophils from SLE patients released IL-33-decorated NETs, further inducing a robust type I IFN response by DCs through their ST2 activation [85]. Conversely, other studies found no statistically significant difference in the serum level of IL-33 between SLE patients and controls [60], or even lower levels in the serum of patients [86,87]. This discrepancy could be attributed to a difference between the detection efficacy of the enzyme-linked immunosorbent assay (ELISA) kits used in the studies, or to the heterogeneity of SLE patient cohorts, especially regarding disease activity or genetic background. 

In contrast, sST2 serum levels have been more consistently reported to be significantly elevated across studies and correlated with the disease activity score (SLEDAI) [21,61,88] and with anti-double-stranded deoxyribonucleic acid (anti-dsDNA) antibodies [60,86]. In addition, Moreau et al. found a statistically significant increase in sST2 serum levels of patients with lupus nephritis compared to SLE patients free of renal involvement [21]. Moreover, sST2 serum levels correlated with urinary proteins in the subgroup of patients with active nephritis [86]. Interestingly, sST2 levels were also found to correlate with proteinuria in immunoglobulin (Ig)A nephropathy, suggesting the hypothetical involvement of sST2 in other kidney disorders [89].

## 5. The Pathophysiological Role of IL-33 in Systemic Lupus Erythematosus

Preclinical studies report conflicting data regarding the role of the IL-33/ST2 axis in the pathophysiology of systemic lupus erythematosus. 

In the lupus-prone model of MRL/Lpr mice, anti-IL-33 treatment from weeks 14 to 20 significantly reduced mortality and lessened serum anti-dsDNA levels and circulating ICs. Renal biopsies showed alleviated renal damage as suggested by the reduced score of glomerulonephritis (GN), reduced renal IC deposition and reduced proteinuria. Finally, anti-IL-33 antibody treatment promoted the expansion of Treg and myeloid-derived suppressor cells (MDSCs) and decreased pro-inflammatory cytokines such as IL-17, IL-1β and IL-6. These data suggest that IL-33 antagonization has a protective effect on SLE [84]. In addition, results from WT mice chronically exposed to IL-33 showed a dramatic increase in B-cell activating factor (BAFF) levels, leading to the production of B and T follicular helper cells, the apparition of germinal centers and the apparition of IgG anti-DNA antibodies. These data suggest the potential involvement of IL-33 as a link between innate and adaptive immunity, and as a potent breaker of immune tolerance through IL-33-mediated BAFF production [88]. 

Conversely, the effect of early IL-33 administration in lupus-prone NZB/W F1 mice from weeks 6 to 12 significantly reduced proteinuria and mortality. Histological analysis revealed a significant reduction in glomerular and tubular damage scores and less deposition of ICs. IL-33 treatment also promoted IgM anti-dsDNA antibodies, IL-10-positive Breg cells and an M2 macrophage gene signature according to RNA sequencing data. These data suggest that IL-33 may exert a protective role during the development of SLE [90]. It has to be underscored that there is an antibody-independent production of B cells relating to autoimmunity, and that lupus mice with B cells unable to produce autoantibodies developed a lessened form of nephritis relative to those without B cells. The mechanisms of the protective role of IgM autoreactive anti-dsDNA antibodies in lupus nephritis are not fully deciphered but could be explained by the significant reduction in the production of pro-inflammatory cytokines such as TNF-α and IFNγ. It has been postulated that, in opposition to IgG anti-dsDNA antibodies harboring a cardinal role in fostering inflammation through the production of inflammatory cytokines by macrophages in the kidneys, IgM antibodies might lessen the inflammatory environment and inhibit the formation of immune complexes [91,92]. In the same line of thought, Stremska et al. studied the effect of IL-233, a hybrid cytokine with active domains of IL-2 and IL-33, as IL-2 and IL-33 have both been shown to expand Treg cells via ST2 ligation [93]. IL233 was shown to induce a durable remission both in established IFN-α-induced lupus GN and in spontaneous GN mouse models (i.e., NZM2328 and MRL/lpr mice). A single course of IL233 daily injections for 5 days induced Treg production and a reduction in pro-inflammatory T cells, without significantly affecting IC deposition within the glomeruli. The induced remission of lupus GN was long-lasting after the treatment [94]. Düster et al. showed a significant reduction in ILC2 in inflamed renal tissue from MRL-lpr mice with GN. After a regiment of two IL-33 injections at 14 and 17 weeks, a significant increase in ILC2 was observed, together with lower scores of lupus nephritis and a decrease in mortality.

In light of the above, one could hypothesize that these contrasting data mirror the dichotomous role of the IL-33/ST2 axis reported in the pathophysiology of inflammatory diseases [95,96]. Depending on the genetic background, the length of the disease (early disease vs. late disease), the immunological background (i.e., pro-inflammatory environment related to an active disease) or the time course of IL-33 treatment (short-term IL-33 depletion vs. long-term IL-33 depletion), the IL-33/ST2 axis could be skewed towards displaying either pro-inflammatory or anti-inflammatory effects. 

The hypothetical involvement of the IL-33/ST2 axis in the pathogenesis of SLE is further detailed in Figure 3. In genetically susceptible subjects exposed to a wide range of environmental factors such as viruses, UV light and stress [3], the products of cell damage arising from injured epithelial barriers lead to the passive release of IL-33. sST2 levels rise in an attempt to counteract the sudden increase in extracellular IL-33, as suggested by immunostaining from patients with lupus nephritis, where an increase in both IL-33 and sST2 was observed [21]. 

In addition, the exposure of numerous self-antigens resulting from dying cells ultimately led to the formation of ICs. Together with products of cell damage, ICs stimulate neutrophils to produce NETs. Once complexed with IL-33, NETs are potent activators of DCs via their ST2 receptor, leading to a potent IFN-α secretion that contributes to the IFN signature of SLE [85]. In addition, functional studies showed that DCs also responded directly to IL-33 through the ST2 receptor and polarized CD4+ T cells into a Th2 phenotype [97]. Chronic exposure of IL-33 induces BAFF secretion by bone marrow stromal cells and possibly other, but not yet identified, radiation-resistant cells that induce B cell proliferation and differentiation into plasma cells, further contributing to germinal center formation, autoantibody production and IC formation [88]. This goes in line with studies performed in lupus patients, where increased serum levels of IL-33 have been shown to correlate with autoantibody levels [58,82]. It must be stressed that the involvement of the IL-33/ST2 axis in the pathogenesis of SLE is very difficult to distinguish from its role, and from the facts shown in other autoimmune diseases.

Under certain, but still to be identified, conditions, possibly in the earlier phase of the disease, the anti-inflammatory effects of IL-33 prevail. Elevated levels of IL-33 induce IL-2 secretion by mast cells and dendritic cells, leading to Treg cell expansion [73,74]. In addition, ST2 has been demonstrated on Treg, Breg and M2 macrophages, leading to anti-inflammatory cytokine production (IL-10, TGF-beta) [70]. Finally, ILC2 and Treg are also a source of AREG, which promotes tissue healing and homeostasis [77]. Together with IFN-related genes, AREG levels were significantly upregulated in PBMCs from SLE patients compared to healthy controls [7]. In a preclinical study, local AREG mRNA expression was significantly increased during the development of LN in a lupus mouse model. In addition, AREG-KO mice experienced significantly more severe scores of lupus nephritis. In vitro assays carried out on CD4+ T cells showed an increased production of pro-inflammatory cytokines such as type I IFN and IL-17A. Treatment of spleen cell cultures with AREG suppressed pro-inflammatory cytokine production and induced the apparition of Treg cells [98].

## 6. Clinical Implications and Future Directions

Current clinical trials using drugs targeting the IL-33/ST2 axis are totally lacking. Even if targeting the IL-33/ST2 axis seems to be a potential therapeutic option according to several preclinical studies in mice [84,90], there is still a long way to go from bench to bedside before currently using IL-33/ST2 therapies in SLE patients. 

At present, several monoclonal antibodies against IL-33 or ST2 are still under development and are currently being tested in phase I and II clinical trials, mainly for the treatment of patients with allergic diseases. In particular, CNTO 7160, a monoclonal antibody against the ST2 receptor, has been investigated in a phase I clinical trial in healthy subjects and patients with asthma or atopic dermatitis [99]. Another selective monoclonal antibody against ST2, astegolimab, showed encouraging results in phase IIb trials for the treatment of severe asthma, with a more striking benefit reported in patients with elevated blood eosinophils [100]. Itepekimab, a monoclonal antibody against IL-33, showed efficacy and safety both as a monotherapy and in combination therapy in patients with moderate to severe asthma [101]. The results of a phase IIa trial suggested that a single dose of etokimab, another monoclonal antibody targeting IL-33, could be effective in desensitizing peanut-allergic patients and in reducing atopy-related symptoms [102]. 

## 7. Conclusions

In this review, the current knowledge about the entanglement of the IL-33/ST2 axis in the pathogenesis of SLE was portrayed. Fundamentally, IL-33 is a pleiotropic molecule but mainly exhibits dual properties of an alarmin, leading to a pro-inflammatory cytokine response and eliciting counteracting homeostatic mechanisms. In SLE, IL-33 exhibits both properties, functioning as a pro-inflammatory alarmin, as well as a promoter of tissue healing and regulatory immune responses. Targeting the IL-33/ST2 axis in SLE could be a potentially interesting therapeutic option in the upcoming years. However, further understanding in deciphering the involvement of IL-33 in SLE is required to better apprehend the conflicting roles of IL-33 in SLE physiopathology and could pave the way for new exciting therapies. 

## Figures and Tables

**Figure 1 ijms-23-03138-f001:**
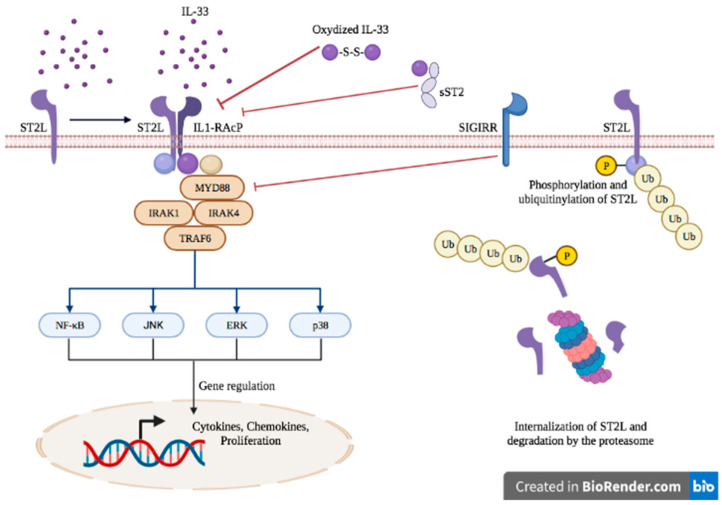
IL-33/ST2 axis signaling.

**Figure 2 ijms-23-03138-f002:**
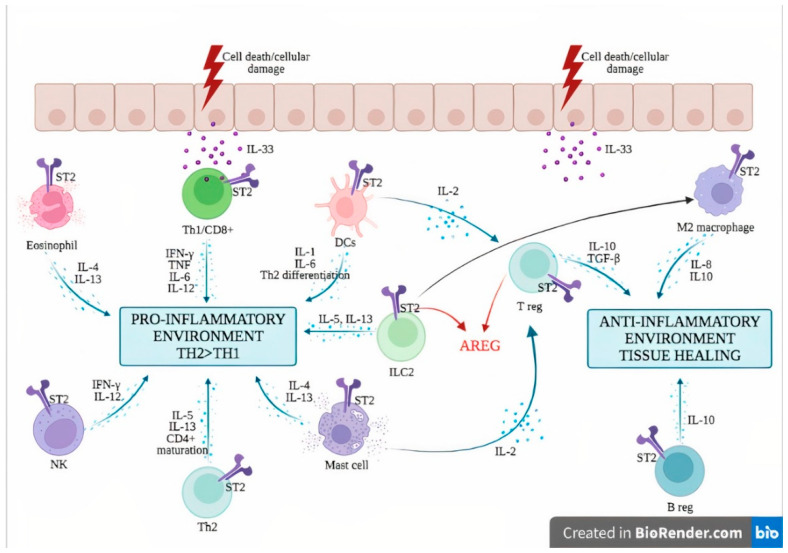
**The dichotomous role of the IL-33/ST2 axis in inflammatory diseases.** IL-33 is passively released by dying cells in the extracellular compartment where it exerts its functions through the receptor ST2. ST2 is found in a wide variety of myeloid and lymphoid cells where it can induce either inflammatory or pro-inflammatory responses depending on the immunological context. The IL-33/ST2 axis induces the secretion of type 2 cytokines such as IL-4 and IL-13 by eosinophils and mast cells, and IL-5 and IL-13 by ILC2 and Th2 cells. IL-33/ST2 mediates the activation of NK cells, leading to the production of IFN-γ and IL-12. Besides the secretion of IL-1 and IL-6, activated DCs induce a Th2 polarization of CD4+ T cells. IL-33/ST2 activation can also activate Th1 cells and CD8+ T cells, leading to type 1 cytokine secretion and cytotoxic activity. On the other hand, IL-33/ST2 also induces IL-2 secretion by mast cells and dendritic cells, leading to Treg expansion. In addition, ST2 has been demonstrated on Treg, Breg and M2 macrophages, leading to anti-inflammatory cytokine production (IL-10, TGF-beta). ILC2 and Treg are also a source of AREG, which promotes tissue healing. Abbreviations: AREG: amphiregulin; Breg: regulatory B cells; DCs: dendritic cells; IFN-γ: interferon gamma; IL-: interleukin; ILC2: innate Lymphoid Cells type 2; NK: natural killer cells; ST2: receptor suppression of tumorigenicity 2; TGF-β: transforming growth factor beta; Th1: type 1 helper cells; Th2: type 2 helper cells; TNF tumor necrosis factor; Treg: regulatory T cells.

**Figure 3 ijms-23-03138-f003:**
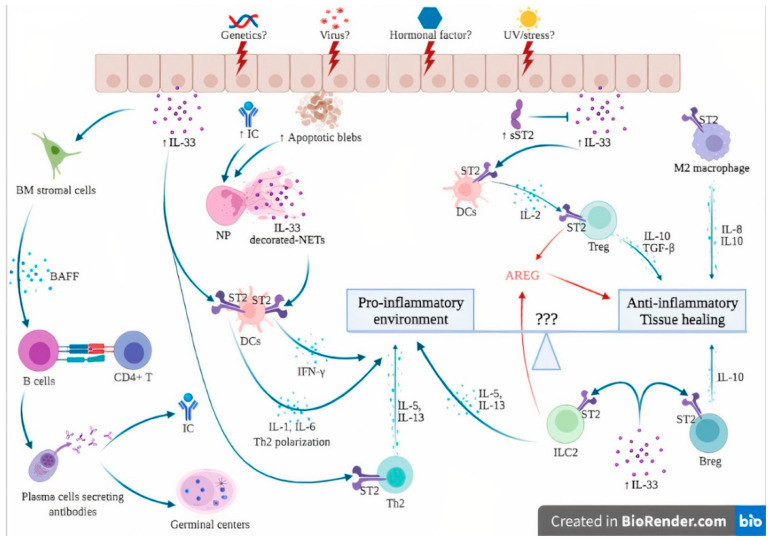
**Hypothesis of the involvement of the IL-33/ST2 axis in the pathogenesis of systemic lupus erythematosus.** The dual role of the IL-33/ST2 axis can be seen as a balance between pro-inflammatory and anti-inflammatory effects. Unknown factors (3 questions marks in the figure) can influence this balance, skewing the immune response toward either pro- or anti-inflammatory states. In genetically susceptible subjects, environmental stimuli such as viruses, UV light and stress may trigger cell death and necrosis of the epithelial barrier, leading to passive release of IL-33, apoptotic blebs and exposure of autoantigens, ultimately leading to the formation of ICs. The products of cell damage, together with ICs, activate neutrophils to produce NETs that complex with IL-33 to activate DCs via their ST2 receptor, leading to a potent type I IFN secretion that contributes to the IFN signature of SLE. In addition, IL-33 also directly activates ST2 expressed by DCs, leading to the Th2 polarization of CD4+ T cells. IL-33 induces BAFF secretion by bone marrow stromal cells and possibly other, but not yet identified, cells that induce B cell differentiation into plasma cells, further contributing to germinal center formation and IC formation. Under certain conditions, probably in the early phase of the disease, the anti-inflammatory effects of IL-33 are dominant. sST2 levels are elevated to counteract IL-33 actions. IL-33/ST2 also induces IL-2 secretion by mast cells and dendritic cells, leading to Treg expansion. In addition, ST2 has been demonstrated on Treg, Breg and M2 macrophages, leading to anti-inflammatory cytokine production (IL-10, TGF-beta). Finally, ILC2 and Treg are also a source of AREG, which promotes tissue healing. Abbreviations: AREG: amphiregulin; BAFF: B-cell activating factor; Breg: regulatory B cells; BM: bone marrow; DCs: dendritic cells; IC: immune complexes; IFN-γ: interferon gamma; IL-: interleukin; ILC2: innate Lymphoid Cells type 2; NETs: neutrophil extracellular traps; NP: neutrophils; sST2: soluble ST2; ST2: receptor suppression of tumorigenicity 2; TGF-β: transforming growth factor beta; Th2: type 2 helper cells; TNFα: tumor necrosis factor alpha; Treg: regulatory T cells; UV: ultraviolet; ⊥: inhibit; blue and red ↓: induce/activate.

## Data Availability

Not applicable.

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
