# Peer review of "Involvement of IL-33 in the Pathophysiology of Systemic Lupus Erythematosus: Review"

_ijms, 2022, doi:10.3390/ijms23063138_

Round 1

Reviewer 1 Report

Authors have performed an interesting and valuable review of literature about IL-33 in the pathogenesis of SLE. However, your manuscript could be developed a bit further. 

Comments.

1

Authors presented extracellular blocking machinery, and intracellular signaling pathway of IL-33 and ST2. Those matters may seem a bit complicated. Additional graphical presentation might be informative.

2

In line 234, authors described IgM anti-dsDNA antibody. The protective effect of IgM antibodies might not be familiar to the readers. So,  additional references and description are required.

3

In line 255-288, authors described the hypothetical contribution of IL-33 in SLE. However, the facts which were proven in SLE, and the facts which were proven only in other diseases are sometimes difficult to distinguish. Authors do well to modify the paragraph.

Author Response

Dear ,

thank you for your remarks 

The manuscript has been updated with the required modifications .

Thanks again

Reviewer 2 Report

This review summarizes the currents knowledge on the potential involvement of the IL-33/ST2 axis in systemic lupus erythematosus (SLE). The Authors provide a detailed update on the current knowledge of biology and functions of the IL-33/ST2 axis in physiologic and pathologic conditions in animal models and humans, especially in inflammatory disorders. Among these, the review is focused on SLE reporting a balanced analysis of the conflicting results reported in the literature on the role played by IL-33 in SLE immunopathogenesis and IL-33 serum levels in SLE patients. The topic is of interest for a better understanding of SLE pathogenesis and for the development of potential future treatment options for SLE patients. The manuscript is well written and can be published in its present form.

Author Response

Thank you very much for your remarks.

Reviewer 3 Report

IL-33 plays an immunomodulatory and pleiotropic role in the immune system. In recent years, the role of IL-33 in the pathogenesis of rheumatic diseases has been confirmed, however, the exact role of the IL-33 / ST2 axis needs to be clarified. The topic is therefore current, though not new. The manuscript presents a comprehensive overview of the data on the subject.

However, in my opinion, the manuscript requires organizing.

The introduction describes the pathogenesis of SLE. Although it does not cover all the knowledge on the subject, it is sufficient to introduce the subject. Nevertheless, a distinction between congenital and acquired immune system disorders in SLE would make the subject easier to understand.

The paragraph ("IL-33 is cytokine ..." verse 60-73) covers the biology and function of IL-33 and some information has been repeated in section 2.

Likewise, some information from section 2 was repeated in section 3.

In section 5, some of the introductory information on the pathogenesis of SLE and other sections are reproduced.

Overall, in all sections, information on IL-33: function, signaling pathway, role in congenital and acquired immunity, including INF gamma release and others is mixed, which makes the subject difficult to read.

I recommend reanalyzing the text, delete unnecessary information and change their order.

Author Response

dear 

Thank you for your remarks . The manuscript has been revamped accordingly but it is very difficult to re organise the text fully ...since the text is already organized in different sub titles so that to enable the reader to follow according to a sequence of events: introduction/biology/ inflammatory diseases and involvement....

We have weeded out accordingly any superficial could text that could hobble the comprehension of the reader.

Hopefully, we hope that suits yours remarks .

Regards

M.S S

Round 2

Reviewer 3 Report

I have no additional comments to the manuscript.